# Managing Post-Transplant Diabetes Mellitus after Kidney Transplantation: Challenges and Advances in Treatment

**DOI:** 10.3390/ph17080987

**Published:** 2024-07-26

**Authors:** Grzegorz Rudzki, Kinga Knop-Chodyła, Zuzanna Piasecka, Anna Kochanowska-Mazurek, Aneta Głaz, Ewelina Wesołek-Bielaska, Magdalena Woźniak

**Affiliations:** 1Department of Endocrinology, Diabetology and Metabolic Diseases, Medical University of Lublin, Jaczewskiego 8, 20-954 Lublin, Poland; maagdalena.wozniak@wp.pl; 2University Clinical Hospital No. 4 in Lublin, Jaczewskiego 8, 20-954 Lublin, Poland; kinga.knop03@gmail.com (K.K.-C.); ewelina_wesolek96@wp.pl (E.W.-B.); 3Saint Queen Jadwiga’s Regional Clinical Hospital No. 2 in Rzeszow, Lwowska 60, 35-301 Rzeszów, Poland; zuz.pia@wp.pl; 4Stefan Cardinal Wyszynski Province Specialist Hospital, al. Kraśnicka 100, 20-718 Lublin, Poland; aniako19@gmail.com; 5Faculty of medicine, Medical University of Lublin, al. Racławickie 1, 20-059 Lublin, Poland; glaz.aneta11@gmail.com

**Keywords:** diabetes mellitus, hyperglycemia, kidney transplantation, insulin resistance, post-transplant diabetes mellitus, new onset diabetes after transplantation

## Abstract

Kidney transplantation is the most effective treatment for end-stage renal failure but is associated with complications, including post-transplant diabetes mellitus (PTDM). It affects the quality of life and survival of patients and the transplanted organ. It can cause complications, including infections and episodes of acute rejection, further threatening graft survival. The prevalence of PTDM, depending on the source, can range from 4 to 30% in transplant patients. This article aims to discuss issues related to diabetes in kidney transplant patients and the latest treatments. Knowledge of the mechanisms of action of immunosuppressive drugs used after transplantation and their effect on carbohydrate metabolism is key to the rapid and effective detection of PTDM. Patient therapy should not only include standard management such as lifestyle modification, insulin therapy or pharmacotherapy based on well-known oral and injection drugs. New opportunities are offered by hypoglycemic drugs still in clinical trials, including glucokinase activators, such as dorzagliatin, ADV-1002401, LY2608204, TMG-123, imeglimine, amycretin and pramlintide. Although many therapeutic options are currently available, PTDM often creates uncertainty about the most appropriate treatment strategy. Therefore, more research is needed to individualize therapeutic plans and monitor these patients.

## 1. Introduction

In patients with end-stage renal disease, kidney transplantation is the most effective and preferred form of treatment, improving the survival of these patients [1]. However, this is associated with a number of possible complications that can negatively affect clinical outcomes. One of these is post-transplant diabetes mellitus (PTDM), which not only affects patients’ quality of life but, more importantly, their survival rate and the viability of the transplanted organ [2].

The aim of the following review is to introduce the issue of diabetes in patients after kidney transplantation and to review the most recent treatments that may provide therapeutic benefits in the indicated group of patients.

Criteria for the diagnosis of PTDM diabetes remain similar to those in the general population and include: 2-h fasting glucose levels ≥ 126 mg/dL (7.0 mmol/L) found in at least two measurements, an adherent glucose level ≥ 200 mg/dL (11.1 mmol/L) with concomitant hyperglycemic symptoms, a 2-h glucose level after a 75-g oral glucose tolerance test (OGTT) ≥ 200 mg/dL (11.1 mmol/L) and an HbA1C level ≥ 6.5% (48.0 mmol/mol) [3].

The first criteria for the classification of diabetes were introduced in 2003 by the American Diabetes Association and the World Health Organization, but they are not perfect and have several limitations [4]. First of all, one of them is a definition that takes into account the level of glycated hemoglobin, HbA1C. It should be noted that based on its measurement, PTDM can be significantly underdiagnosed due to anemia, which is a common problem in patients with chronic kidney disease or shortened red blood cell survival in transplant patients. It is recommended that the diagnosis of PTDM diabetes not be made solely on the basis of this parameter until 12 months after transplantation [5,6].

The definition of post-transplant diabetes is often ambiguous and controversial. The term new onset diabetes after transplantation (NODAT) refers to metabolic changes that occur after transplantation and, importantly, require the direct exclusion of carbohydrate disorders before transplantation [7]. At an expert meeting in Vienna in 2013, it was proposed to replace the term NODAT with post-transplant diabetes mellitus (PTDM). This change seems to show the differentiation of the indicated group of patients, in whom diabetes may have been undiagnosed before transplantation or developed independently after transplantation [8]. Furthermore, the American Diabetes Association guidelines have adopted the concept of post-organ transplant PTDM diabetes, which was previously classified as “other non-specific types of diabetes” [9]. Due to the above stipulations and consensus, the term used by the authors of this paper will be post-transplant diabetes mellitus (PTDM), although the term NODAT is still encountered in the literature.

Understanding epidemiology and risk factors is essential for optimizing care and planning intervention strategies. Epidemiological data on the prevalence of diabetes mellitus after kidney transplantation are inconclusive. This is likely due to the lack of a clear definition of PTDM and the frequent attribution of this group of patients to classic type 2 diabetes mellitus (DMT2). Some epidemiological data suggest that the prevalence ranges from 4 to 25% [10]. Other studies indicate that post-transplant diabetes and pre-diabetes may involve up to 20–30% of patients [11,12]. On the contrary, it is known that PTDM most frequently develops between 3 and 6 months after transplantation and is often classified as early onset. This early post-operative period is critical due to factors such as surgical stress, high doses of corticosteroids and the initiation of immunosuppressive therapies like calcineurin inhibitors, which contribute to hyperglycemia and the onset of diabetes. The late onset of PTDM, when it develops after 12 months from the transplant, is comparable to the incidence of classic DMT2 in the general population. Studies have indicated that the annual incidence of late-onset PTDM is around 7%, which aligns with the prevalence of DMT2 in the general population. [13] However, noteworthy is the fact that no cut-off period exists for the diagnosis of PTDM, which additionally poses a diagnostic challenge. The impact of timing on the diagnosis of post-transplant diabetes has yet to be established [1].

The pathophysiology of PTDM development resembles that of DMT2, although it is based on other risk factors, making this diabetes different in nature and course from classic DMT2. The primary mechanism of PTDM is based on increased insulin resistance, impaired insulin production as a result of pancreatic β-cell damage and uncontrolled glucagon release [14]. Some researchers have suggested that insulin resistance is the principal mechanism for the development of diabetes after transplantation [15], while others have argued that impaired pancreatic β-cell function is the main cause of the disease [16,17]. It is certain that pancreatic β-cells undergo a process of dedifferentiation, changing their form from highly specialized to primitive, under the influence of inflammatory factors, oxidative stress, mitochondrial dysfunction and other stress-critical transcription factors [18]. Often, impaired pancreatic β-cell function is observed even before transplantation. Up to 80% of kidney recipients develop elevated levels of proinsulin-indicating β-cell dysfunction and increasing insulin resistance before qualifying for transplantation [16].

One of the key reasons for the development of insulin resistance in PTDM is immunosuppressive treatment [19]. Cyclosporine and tacrolimus contribute to the removal of the GLUT4 receptor from the surface of adipocytes, thereby inhibiting insulin-independent glucose uptake [20]. Calcineurin inhibitors, particularly tacrolimus, show deleterious effects on pancreatic β-cells, leading to their apoptosis and thus reduced insulin secretion, but also the induction of insulin resistance [21]. Such effects, although of somewhat lesser intensity, are also exerted by cyclosporine [22]. The calcineurin inhibitor additionally affects muscle fibers by transcribing genes that favor insulin-resistant fast myosin fibers [23].

Insulin resistance and impaired pancreatic β-cell function are the main pathophysiological causes of PTDM development, to which certain risk factors are predisposed. They are summarized collectively in Table 1 [24,25].

## 2. Methods

This systematic review aims to summarize the latest literature regarding the latest treatment of diabetes in renal transplant patients. The current literature was reviewed by searching for publications from the period between 2014 and 2024 using the search phrases “New Onset Diabetes After Transplantation” AND “Post-Transplant Diabetes Mellitus” AND “kidney transplantation” in the PubMed database. Three authors independently assessed the eligibility and quality of the study and performed data extraction. All selected studies were included if the study population consisted of adult renal transplant patients diagnosed with impaired glucose tolerance, diabetes mellitus (DM), new onset post-transplant diabetes (NODAT) or post-transplant diabetes mellitus (PTDM). The exclusion criteria were as follows: articles available in a language other than Polish or English, as well as other publication types such as editorials, reviews, posters and letters.

## 3. Basic Medications in the Post-Transplant Period and in the Development of Diabetes after Transplantation

### 3.1. Treatment Used after Transplantation 

Immunosuppressive treatment after organ transplantation involves the simultaneous use of several drugs in specific regimens, depending on the type of transplanted organ, the degree of immune risk, the severity of metabolic disorders, the presence of comorbidities and the function of the transplant. Its purpose is to inhibit or reduce the recipient’s immune response to transplant antigens. In addition to its beneficial effects, it can contribute to infections, tumors and the appearance of toxic effects against other tissues and organs as well [26,27].

The choice of an appropriate treatment regimen depends on a number of factors, including donor- and recipient-dependent prognostic factors, the interaction of the chosen drug with other drugs already in use, the cost of treatment and expected side effects [28].

#### 3.1.1. Glucocorticosteroids

For 50 years, glucocorticosteroids (GCSs) have been the standard for immunosuppression. They act non-specifically on the immune system, interfering with the NF-kappaB pathway, reducing cytokine secretion and affecting the function of all immune cells [29]. The advantage of using these drugs is undoubtedly the inhibition of cellular and humoral responses achieved by blocking the transcription of genes responsible for producing the inflammatory response, including a reduction in the production of the interleukins IL-1, IL-2, IL-6, interferon gamma and tumor necrosis factor (TNF-alpha) [30]. While chronic steroid therapy is associated with side effects that include diabetes, hypertension, emotional vacillation, swelling, impaired wound healing and susceptibility to infection, it is a crucial treatment for transplant patients. It is important to emphasize that by increasing gluconeogenesis and hyperglycemic effects, they increase the incidence of obesity or insulin resistance, risk factors that can lead to carbohydrate metabolism disorders [31].

GCSs are the agents with the highest diabetogenic potential, which depends on the drug dose administered [32]. In a study involving 25,837 kidney transplant recipients without diabetes, the incidence of PTDM within three years of transplantation was 16.2% and 17.7% when treated with maintenance steroids. Patients who received immunosuppressive treatment regimens containing steroids had a 42% higher risk of PTDM in comparison to those not receiving steroid therapy, indicating a strong association between steroid use and PTDM [33].

Additionally, it has been observed that the rapid discontinuation of these drugs and immunosuppressive strategies that limit them can significantly reduce the incidence of not only hyperglycemia but also dyslipidemia, hypertension, osteoporosis or cardiovascular complications [34].

In a study by Pahor et al. aimed at finding possible associations between post-transplant diabetes mellitus and polymorphisms of the glucocorticoid pathway, it was observed that the presence of GSTP1 (Glutathione S-transferase P1) genotypes was associated with a higher likelihood of PTDM. Importantly, these polymorphisms can already be determined prior to kidney transplantation and, when assessed in daily practice, could help plan the early withdrawal of GCSs after transplantation [35].

#### 3.1.2. Immunosuppressive Drugs

Calcineurin inhibitors (CNIs) are medications used in immunosuppressive treatment regimens for solid organ transplants. Their action consists of inhibiting calcineurin phosphatase, leading to a reduction in T-lymphocyte activation and inhibition of the release of inflammatory cytokines. A number of side effects can occur, in particular acute and chronic nephrotoxicity, electrolyte disturbances, metabolic acidosis and neurotoxicity, which may contribute to the development of PTDM [36].

Tacrolimus has been proven to be associated with a higher incidence of PTDM, and hypomagnesemia, which is also more common with the use of this drug, has been shown to be an independent risk factor for the development of PTDM [37] and has also been shown to have more frequent hyperglycemic effects compared to cyclosporine [14].

A study by Terrec et al. evaluated the benefits of switching from CNI-based immunosuppression to belatacept-based immunosuppression in renal transplant patients with coexisting diabetes. This change significantly improved glycemic parameters. In renal transplant recipients, HbA1c values decreased from 7.2 ± 1 to 6.5 ± 1% (*p* = 0.001). Furthermore, HbA1c values decreased significantly regardless of whether diabetes was controlled at the time of study inclusion or not (HbA1c ≤ 7% or >7%) [38].

Another group of immunosuppressive drugs used in kidney transplant patients are mTOR inhibitors, which include everolimus and sirolimus. Their mechanism of action involves binding to the cytosolic protein FKPB-12, by which they inactivate mTOR kinase, leading to the inhibition of lymphocyte activation and immune reactions. Although they are thought to have diabetogenic effects, their use does not significantly increase the incidence of PTDM [39]. It has been shown that sirolimus can cause β-cell failure due to its mechanism of mTORC1-mediated inhibition of cell proliferation and survival and the disruption of mTORC2 signaling, thereby impairing insulin sensitivity [40]. In contrast, Montero et al., in a meta-analysis evaluating the association of CNI with mTOR inhibitors in de novo renal transplant recipients, observed no increase in annual PTDM compared with CNI and antiproliferative drugs in 13 studies [41]. There is no evidence to suggest a risk of increased glycemia with antiproliferative drugs such as mycophenolate mofetil or azathioprine [39].

### 3.2. Treatment of PTDM

The treatment of PTDM is challenging due to the higher risk of serious side effects in transplant recipients with impaired renal function. Different types of drugs are used, differing in mechanism of action and side effect profile. There are also differences in their indications depending on variables such as history of previous diabetes, renal function, allograft function and immunosuppressive drugs taken. Pharmacotherapy for PTDM is constantly evolving [14,42].

#### 3.2.1. Lifestyle Modification

Patients awaiting kidney transplantation should undergo a thorough evaluation to detect risk factors for post-transplant diabetes. Family history is also not without significance. The early identification of patients at risk allows for the implementation of appropriate preventive management [43]. In the prevention of PTDM, moderate to vigorous physical activity for at least 150 minutes per week is recommended according to guidelines for the general population. However, specific recommendations relating to transplant recipients are lacking [44]. The introduction of a Mediterranean diet based on whole grains, legumes, fruits, vegetables, olive oil and fish, with limited consumption of dairy products and meat, may also be beneficial. Such nutrition can improve insulin sensitivity and pancreatic β-cell function. [45]. Studies that have extensively described the properties of Mentha (M.) species plants should also be mentioned. The essential oils of M. virdis, M. arvensis L and M. suaveolens were found to contribute to the inhibition of α-glucosidase and α-amylase. An extract prepared from the leaves of M. spicata also shows antidiabetic effects and may contribute to a reduction in cholesterol, low-density lipoproteins and triglycerides. However, more research is required on the reduction of blood sugar levels when using the extract of these plants [46,47].

Kuningas et al. evaluated the benefits of lifestyle modification in kidney transplant recipients. There was no significant improvement in glucose metabolism 6 months after the change in habits, but there was a reduction in body weight, fat mass and the incidence of post-transplant diabetes mellitus (7.6% with active intervention versus 15.6% with passive lifestyle intervention; *p* = 0.123) [48].

Although lifestyle modification can improve glycemic control and has its uses as a strategy to prevent PTDM, overt post-transplant diabetes mellitus tends to have a more acute onset than DMT2 and most often requires early pharmacological treatment rather than relying solely on lifestyle change interventions to reduce hyperglycemia [49].

#### 3.2.2. Insulin Therapy

Insulin therapy remains the primary treatment for PTDM [50]. It plays a particularly important role in the early post-transplant period (0–45 days). Insulin is the only hypoglycemic drug that can be used for the first 7 days after surgery [51]. Some studies have suggested that the early administration of basal insulin may be effective in preventing the development of PTDM in patients with hyperglycemia in the early postoperative period and contribute to improving pancreatic β-cell function [50,52]. On the other hand, in the case of insulin resistance induced by the use of glucocorticosteroids in transplant recipients, the administration of long-acting basal insulin alone is not an appropriate approach. The optimal approach in such a case may be a combination of long-acting and rapid-acting insulin. Then, the total daily insulin dose is approximately divided into 30% long-acting insulin and 70% rapid-acting insulin [42].

#### 3.2.3. Metformin

Metformin reduces hepatic glucose production by phosphorylating the transcriptional response element binding protein cAMP (CREB), reducing the expression of genes that induce gluconeogenesis [53]. Metformin may offer some advantages over other glucose-lowering drugs, particularly with regard to its low risk of hypoglycemia and neutral effect on body weight [51,54]. Other benefits of metformin include its low cost, cardioprotective effect and few interactions with other medications. However, the administration of metformin can be controversial, which has to do with the risk of acute kidney injury (AKI) and lactic acidosis while taking the drug, especially when the estimated glomerular filtration rate (eGFR) is <30 mL/min/1.73 m^2^. The risk of lactic acidosis is greatest in the early period after kidney transplantation [53]. According to the Food and Drug Administration (FDA), metformin can currently be implemented in patients with an eGFR of 60 to 45 ml/min and is allowed to continue in patients with an eGFR of 45 to 30 mL/min. If the eGFR is less than 30 ml/min, metformin is contraindicated [45,55,56].

#### 3.2.4. Thiazolidinediones

Thiazolidinediones, such as pioglitazone, are also proving to be effective pharmacotherapies. Drugs in this group increase insulin sensitivity in peripheral tissues through the activation of PPARγ receptors [57]. Nevertheless, their use is associated with higher cardiovascular risk, a major limitation, especially in transplant recipients [53]. Thiazolidinediones can cause side effects such as edema, congestive heart failure, weight gain and an increased risk of fractures [50]. Given that the baseline risk of developing all these complications is already very high in transplant patients, thiazolidinediones should not be prescribed in this particular group of patients [55].

#### 3.2.5. Sulfonylurea Derivatives

Sulfonylurea derivatives are divided into first- and second-generation drugs. First-generation drugs include talbutamide, while second-generation drugs include glibenclamide, glycoside, glipizide, gliquidone and glimepiride. These groups differ in structure. The agents lower blood glucose levels by increasing insulin secretion from pancreatic β-cells. First-generation sulfonylurea derivatives have been largely replaced in routine use by second-generation agents, which are more potent and given in smaller doses, mostly once daily [58].

These drugs induce insulin secretion by binding to the ATP-sensitive potassium channel (KATP), which is composed of four subunits of the sulfonylurea receptor SUR1, ABCC8 and four subunits of the Kir rectifier potassium channel. Binding of the sulfonylurea leads to the closure of the KATP channel, which leads to a change in the resting potential of the cell. This causes calcium influx and stimulation of insulin secretion [59].

Extreme caution should be exercised during the treatment of sulfonylurea derivatives in organ transplant patients due to the risk of hypoglycemia, especially in those treated with cyclosporine A, as active metabolites of sulfonylurea derivatives may accumulate during use in transplant patients. Severe hypoglycemia can occur with these medications as a result of overdose or drug interactions with azole antifungals or other inhibitors of cytochrome P2C9, responsible for the metabolism of sulfonylurea derivatives. Glipizide is preferred only in limited amounts, since <5% of the drug is excreted unchanged by the kidneys. This proves that even glipizide should be avoided in patients with severe disease to avoid hypoglycemia. No change in cyclosporine levels was found in kidney transplant recipients with PTDM during glipizide administration. Drugs in this group also increase the risk of cardiovascular complications, especially in obese patients and those with a history of vascular disease. In patients with PTDM, especially at the onset of the disease, pancreatic β-cells can be severely damaged by immunosuppressive agents, which raises an additional concern about the use of sulfonylurea derivatives [39,60].

#### 3.2.6. DPP-4 Inhibitors

Dipeptidyl peptidase-4 (DPP-4) inhibitors may be another group of drugs administered. Their mechanism of action involves inhibiting an enzyme that degrades two gut-derived incretin hormones, glucagon-like peptide 1 (GLP-1) and glucose-dependent insulinotropic peptide (GIP). Thus, DPP-4 inhibitors stimulate insulin secretion and reduce glucagon secretion in a glucose-dependent manner. Both effects contribute to a blood glucose-lowering effect [45]. A study by Haidinger et al. showed that DPP-4 inhibitors are well tolerated, safe and effective in improving glucose metabolism in transplant recipients. Beneficial changes include reductions in fasting glucose and glucose levels (2HPG) in the 2-hour oral glucose tolerance test (OGTT) (vildagliptin: 2HPG = 182.7 mg/dL; placebo: 2HPG = 231.2 mg/dL; *p* ≤ 0.05), as well as a reduction in HbA1c (vildagliptin: HbA1c = 6.1%; placebo: HbA1c = 6.5%; *p* ≤ 0.05). DPP-4 inhibitors have a more pronounced effect on lowering plasma glucose levels in the OGTT test, which may be related to the mechanism of action of these drugs, which is based on a selective increase in insulin secretion, while patients with PTDM have a defect in the form of the reduced secretion of this hormone. In addition, drugs in this group show a low risk of hypoglycemia [61].

Given that the risk of PTDM is greatest in the initial post-transplant period, Santos et al. tried to assess whether the early administration of DPP-4 inhibitors within the first week after kidney transplantation in patients with hyperglycemia could prevent the development of PTDM. However, the results did not show statistical significance. Patients treated with sitagliptin in the early post-transplant period had a 2HPG of 174.38 ± 77.93 mg/dL 3 months after discontinuing the drug, while those in the placebo group had 171.86 ± 83.69 mg/dL (*p* = 0.918). Thus, the beneficial effect of DPP-4 inhibitors in this regard was not confirmed [62]. In conclusion, the advantages of DPP-4 inhibitors may include the absence of hypoglycemic episodes, ease of use and good tolerability by patients with minimal side effects [63].

#### 3.2.7. SGLT-2 Inhibitors

Sodium-glucose cotransporter-2 (SGLT-2) inhibitors are another group of medications that can be administered to patients with PTDM. They act selectively on the sodium-glucose cotransporter 2 in the proximal tubule of the nephron, which reabsorbs about 90% of filtered glucose. Blocking glucose reabsorption results in glucosuria, osmotic diuresis and natriuresis, leading to a reduction in blood glucose concentrations without stimulating insulin release [52]. Studies have suggested that drugs in this group lower HbA1c levels and lead to weight loss in patients with PTDM. In doing so, they do not cause serious side effects such as euglycemic ketoacidosis. There are also no significant changes in creatinine levels. Despite the many advantages and positive aspects of the use of SGLT-2 inhibitors, their effect is strongly correlated with the eGFR value; the amount of HbA1c reduction depends on the eGFR and the baseline HbA1c level. SGLT-2 inhibitors are not as effective in transplant recipients compared to other diabetic groups, which may be due to the lower eGFR value in this group of patients. In patients with chronic kidney disease (CKD) after transplantation, achieving high eGFR values is a major challenge. When empagliflozin was used in patients with eGFR < 50 ml/min/1.73 m², glucosuria and HbA1c reduction were negligible. Due to this limitation, the administration of SGLT-2 in monotherapy is not preferred [64,65].

Another interesting aspect remains the mild increase in blood ketones observed in patients receiving SGLT-2 inhibitors. Under such conditions, β-hydroxybutyrate is freely taken up by various organs and oxidized to fatty acids. In kidney transplant patients, such a change in metabolism and its increase at the mitochondrial level may improve the condition of the transplanted organ and have a beneficial effect on immune cells, especially T lymphocytes [65,66].

Post-transplant erythrocytosis (PTE) has been observed in kidney transplant patients, occurring in 8–15% of recipients and affecting patients with well-preserved graft function, usually 8–24 months after surgery [64]. Erythocytosis has also been observed in patients receiving SGLT-2 inhibitors, in which case it usually resolves after discontinuation of the medication. The mechanism of SGLT-2 inhibitor-induced erythocytosis has not yet been fully elucidated [67].

#### 3.2.8. GLP-1 Analogs

GLP-1 (glucagon-like peptide 1) analogs are incretin drugs that stimulate glucose-dependent insulin secretion, inhibit glucagon synthesis, slow gastric emptying and enhance satiety [68]. These medications reduce the incidence of cardiovascular events and all-cause mortality, as well as the progression of kidney disease in DMT2, and although they cause gastrointestinal side effects such as nausea, vomiting and diarrhea in some patients, they can be very useful in overweight or obese patients [51]. Studies have suggested that the use of GLP-1 analogs can reduce the need for exogenous insulin, a treatment that appears to be safe and does not increase the risk of allograft failure [43]. A study by Halden et al. showed how GLP-1 analogs can affect the pathophysiological defects found in patients with PTDM, namely reduced glucose-induced insulin secretion and impaired glucagon suppression. These drugs appear to normalize these abnormalities [69]. Compared to SGLT-2 inhibitors, GLP-1 analogs result in lower HbA1c levels, regardless of renal function. However, the retrospective study by Liou et al. illustrated that this involved patients with diabetes and uncontrolled hyperglycemia without clearly distinguishing whether it was classic DMT2 or diabetes that emerged only after transplantation [70]. Collectively, the effects of SGLT-2 inhibitors and GLP-1 analogs in organ transplant patients are summarized in Table 2 [70,71,72,73,74,75,76,77,78,79].

#### 3.2.9. Long-Acting Dual GIP and GLP-1 Receptor Agonist

Tirzepatide is a new therapeutic compound with proven efficacy in maintaining normal glycemia in patients with DMT2. This polypeptide is a dual GIP and GLP-1 receptor agonist [80,81]. Its mechanism of action involves activation of the GLP-1 signaling pathway, which mobilizes glucose-dependent insulin secretion through GIP receptor activity. The drug also has a long half-life, so the time during which blood glucose levels remain within a safe range—71–140 mg/dL—is extended [82,83]. In addition, it increases the sensitivity of pancreatic β-cells to glucose, due to the fact that it enhances first- and second-phase secretion in a glucose-dependent manner. The efficacy of tirzepatide has been proven in five phase III clinical trials—SURPASS 1–5. In all of these trials, the hyperglycemic results obtained in the tirzepatide treatment groups were more favorable than those of patients in the reference groups. Moreover, the drug had an effect on reducing patients’ body weight. Tirzepatide also showed beneficial effects on the cardiovascular system by causing a reduction in the endothelial damage marker ICAM-1 (Intercellular Adhesion Molecule 1), the proinflammatory cytokine YKL-40 (chitinase-3-like protein 1) and blood pressure [84]. In the SURPASS-4 trial, tirzepatide was shown to have an effect that almost doubles the risk of renal failure compared to insulin glargine [85,86]. The drug, especially when administered with insulin-stimulating drugs, may slightly increase the risk of hypoglycemia. Depending on the edition of the SURPASS study, the percentage of hypoglycemia occurrence ranged from 0% to 19.3% [87].

### 3.3. New Directions in the Treatment of Diabetes Mellitus

Despite the known mechanisms of action of the mentioned drugs and their convenient availability, a significant proportion of patients with PTDM do not achieve the desired therapeutic goals. This prompts the search for new therapeutic pathways. The following are drugs that may represent chances for new therapies for the treatment of diabetes.

#### 3.3.1. Glucokinase Activators

Glucokinase activators (GKAs) are a class of antidiabetic drugs developed to regulate blood glucose levels and improve β-cell function in diabetic patients. Glucokinase, also known as hexokinase IV, is a key enzyme present mainly in the liver and pancreatic β-cells. In β-cells, it acts as a glucose sensor, initiating glucose phosphorylation, which leads to ATP production and inhibition of ATP-sensitive K+ channels. This, in turn, opens calcium channels, promoting insulin release. In the liver, glucokinase acts as a “gatekeeper” for cells, where glucose phosphorylation culminates in glycogen synthesis [88].

##### Dorzagliatin

Dorzagliatin, a dual-acting GKA, targets both the liver and pancreas and has successfully completed two phase III trials, demonstrating favorable results in diabetes treatment. Dorzagliatin is being studied for its potential to improve renal function in patients with DMT2 and early kidney damage. It is an interventional study in which dorzagliatin is administered together with metformin and is designed to evaluate the effects of this drug on various indices of renal function, such as eGFR, creatinine levels, cystatin C and TNF-1. The results of this study may provide important information on the safety and efficacy of dorzagliatin in the context of renal function [89].

Although dorzagliatin has not previously been extensively studied for its direct effects on the kidneys, its ability to activate glucokinase may offer benefits in regulating blood glucose levels without adversely affecting kidney function. This is crucial, especially in the context of potential treatment for people with post-transplant diabetes mellitus, where preserving kidney function is a priority. If studies confirm that dorzagliatin improves kidney function, or at least does not worsen it in diabetic patients, it could be a promising candidate for administration in kidney transplant patients who develop PTDM [90].

##### TMG-123

TMG-123 is an innovative glucokinase activator that is being investigated for the treatment of DMT2. Based on available information, TMG-123 has potentially long-lasting effects in lowering glucose levels without adversely affecting liver and plasma triglyceride levels, even after chronic treatment. Such effects make it a promising candidate for patients who need effective glucose control without the added risk of dyslipidemia, a common problem among existing diabetes therapies. TMG-123 is also being studied in combination with metformin, suggesting potential synergistic effects in diabetes management, especially in terms of improving glucose tolerance and lowering HbA1c levels in animal models, which may translate into benefits for humans with DMT2. Unfortunately, no detailed information is currently available on the effects of TMG-123 on renal function [91,92,93].

##### ADV-1002401

A similar drug from the group of glucokinase activators being studied for the treatment of DMT2 is ADV-1002401. Currently, there is no detailed information on how it is metabolized or how it directly affects kidney function. The drug is currently being tested in a first-in-human study, where its safety, tolerability, pharmacokinetics and pharmacodynamics are being evaluated. The results of these studies will provide further information on the potential efficacy and safety of ADV-1002401 in the treatment of DMT2 [88].

##### LY2608204

Another medication in this group is LY2608204, which is currently being studied by Eli Lilly in collaboration with Yabao Pharmaceuticals. Another is globalagliatin, on which studies are currently underway [94,95].

#### 3.3.2. Imeglimin

The investigational drug imeglimin, representing a new pharmaceutical group called “glimin”, stands out for its innovative mechanism of action compared to other medications. It positively affects insulin sensitivity and reverses pancreatic β-cell dysfunction. Studies have confirmed that imeglimin exhibits potent anti-diabetic effects, normalizing glucose homeostasis by acting on various metabolic pathways. The key effects of imeglimin include improved insulin sensitivity through potential effects on glucose transporter-4 (Glut-4) and insulin receptor autophosphorylation. In addition, imeglimin has antiapoptotic effects, stimulating pancreatic β-cell function in patients with DMT2. It also reduces hepatic gluconeogenesis by reducing phosphoenolpyruvate and glucose-6-phosphatase activities in hepatocytes, while improving mitochondrial function and regulating intracellular energy production. Imeglimin is also important for pancreatic function, stimulating insulin secretion and increasing cellular Nicotinamide Adenine Dinucleotide (NAD+) levels and calcium ion mobilization, resulting in better insulin exocytosis efficiency. Moreover, imeglimin exhibits antioxidant activity by inhibiting the production of reactive oxygen species in the mitochondria. The pharmacokinetics of imeglimin suggest efficacy in lowering glycated hemoglobin (HbA1c) levels within 16 weeks of treatment. The drug presents an innovative approach to treating DMT2, offering prospects for improving insulin sensitivity, mitochondrial function and reducing oxidative stress. Launched in Japan under the trade name Twymeeg, imeglimin offers a promising safety and efficacy profile, as evidenced by phase II and III clinical trials showing moderate reductions in HbA1c and good patient tolerability. Further studies are needed to obtain more detailed data on imeglimin’s mechanism of action and safety profile, which are currently the subjects of ongoing clinical evaluations [96,97,98,99,100].

#### 3.3.3. Amycretin 

Amycretin is a new diabetes drug currently under development by Novo Nordisk. This drug has just completed phase 1 of clinical trials, and the company plans to initiate phase 2 in the second half of 2024. The preliminary study results are very encouraging. During the phase 1 clinical trials, participants lost an average of 13.1% of their body weight after 12 weeks of drug use, surpassing results from other popular fat-reducing drugs. Although information on the clinical details and mechanism of action of amycretin is still limited, its development indicates potential efficacy in glucose control and effects on weight reduction, which is important for people with DMT2 battling obesity. If the results of the next phases of research confirm these initial observations, amycretin could become an important tool in the management of diabetes and related health conditions. Currently, there is a lack of detailed information on the effects of amycretin on renal function in available sources [101,102,103,104].

#### 3.3.4. Pramlintide

Pramlintide is an analog of amylin, a hormone produced by the β-cells of the pancreas that works with insulin to regulate blood glucose levels, especially after meals. It is an antidiabetic drug used to treat DMT1 and DMT2 in patients who also use insulin for meals and have problems with blood glucose control. Pramlintide works by slowing gastric emptying, which helps control glucose levels after a meal. A study by Ruchi et al. showed that pramlintide effectively reduces post-meal glucagon concentrations by inhibiting glucagon secretion without affecting glucagon clearance. It also increases the feeling of satiety, which can help reduce body weight. This is important because excessive glucagon secretion after meals is one of the mechanisms leading to hyperglycemia in patients with type 1 diabetes. Pramlintide is administered as an injection (subcutaneous injection) and is usually administered in combination with insulin before meals. It is important that patients carefully monitor their blood glucose levels to avoid hypoglycemia, especially at the beginning of treatment. Pramlintide is metabolized mainly by the kidneys and excreted from the body through this route. The drug undergoes little or no metabolism in the liver, which means that its elimination is mainly via the kidneys.

After a single subcutaneous injection, pramlintide concentrations peak in plasma after about 20 min, regardless of dose, and then decrease over the next 3 h. The plasma half-life of pramlintide is approximately 50 minutes. Pramlintide is an effective and safe adjunct to insulin therapy. By inhibiting glucagon secretion after meals and improving overall glycemic control, pramlintide can significantly improve patients’ quality of life. Current study results support its potential as an important component of type 1 diabetes therapy [105,106].

As with many new drugs in development, data on their specific effects on various body systems, such as the kidneys, may be limited to the results of initial clinical trials, which focus primarily on the overall efficacy and safety of the drug.

The aforementioned medications are still in various phases of clinical trials and may offer significant improvements in the treatment of diabetes, especially in terms of long-term glucose control and their potential impact on diabetes-related complications. Obviously, each of these drugs must undergo rigorous testing to verify their efficacy and safety before they are approved for broader use.

## 4. Summary and Conclusions

Understanding the epidemiology and risk mechanisms of PTDM is fundamental to improving the care of kidney transplant patients. This paper highlights the diagnostic difficulties of PTDM, current therapeutic options and new hypoglycemic drugs undergoing clinical trials. Incorporating these new drugs into diabetes management protocols could offer novel therapeutic pathways for patients with diabetes, addressing specific aspects of the condition such as glucose control, weight management and potential renal benefits. Continued research and clinical evaluation are essential to ascertain their efficacy, safety profiles and optimal placement within diabetes treatment strategies. In this review, consideration was given not only to the glycemic effects of available therapies, but also to possible effects on graft function. It was shown that PTDM still remains a carbohydrate disorder that is not well understood and may carry both diagnostic and therapeutic difficulties for practicing physicians. With this in mind, as well as risk factors affecting graft survival and the impact of transplantation on patients’ quality of life, further research should focus on developing individualized strategies for the diagnosis, treatment and monitoring of patients with PTDM.

## Figures and Tables

**Table 1 pharmaceuticals-17-00987-t001:** Non-modifiable and modifiable risk factors for diabetes after transplantation.

Figure	Risk Factors	Description
non-modifiable	Age	2.9-fold higher risk in patients over 45 years of age
	Origins	African American, Hispanic, South Asian descent
	Deceased donor	
	Previous glucose intolerance	e.g., during pregnancy, steroid therapy
	Male donor	
	Genetic	association with HLA B27, HLA A28, HLA A30, HLA Bw42, adiponectin 276G/T, KCNQ1, NFATc4
modifiable	Obesity	higher risk 1.73 times
	Sedentary lifestyle	
	Metabolic syndrome	
	Viral infections	e.g., HCV, CMV
	Corticosteroids	
	Calcineurin-inhibitors	
	Sirolimus	
	Acute rejection	

**Table 2 pharmaceuticals-17-00987-t002:** Research summary of antidiabetic drugs in solid organ transplant patients with post-transplant diabetes mellitus (PTDM).

Author	Transplanted Organ	N Patients	Agent	HbA1c	Oral Glucose Insulin Sensitivity (OGIS) Index	Body Weight	eGFR	Other
**SGLT2**								
Schwaiger et al. [71]	kidney	14 (4 weeks) +8 (12 months)	Empagliflozin	increase from 6.5 ± 0.8% to 6.6 ± 0.7% (*p* = 0.12)	decrease from 390 ± 66 to 328 ± 85 mL/min per m^2^ (*p* = 0.01)	decrease of 1.6 kg	decrease from 55.6 ± 20.3 to 47.5 ± 15.1 mL/min per 1.73 m^2^ (*p* = 0.008) (after 4 weeks)	OGTT increased from 232 ± 82 mg/dL to 273 ± 116 mg/dL (after 4 weeks, *p* = 0.06) and to 251 ± 71 mg/dL (after 12 months, *p* = 0.41); beta cell glucose sensitivity improved from 28.6 ± 17.1 to 36.6 ± 23.5 pmol·min^−1^·m^−2^·mM^−1^ (*p* = 0.06).
Halden et al. [72]	kidney	N = 22 vs. n = 22 (placebo)	Empagliflozin	decrease from 0.2% to 0.1%	there was no significant difference in insulin sensitivity index during treatment (*p* = 0.59)	decrease of 2.5 kg (*p* = 0.014)	there was no significant difference after 24 weeks	Patients with eGFR ≥ 60 mL/min/1.73 m^2^ showed a trend towards greater reductions in HbA1c compared with patients with eGFR < 60 mL/min/1.73 m^2^; Insulin secretion did not change significantly; C-peptide concentration after 2 hours increased in the empagliflozin group compared with a decrease in C-peptide concentration in the placebo group; 24-h renal glucose excretion increased; Renal glucose excretion decreased with worsening renal function. There were no significant differences in weight reduction between patients with baseline eGFR > 60 mL/min/1.73 m^2^ and eGFR < 60 mL/min/1.73 m^2^ (*p* = 0.97).
Mahling et al. [73]	kidney	10	Empagliflozin	decrease from 7.3% to 7.1%	no data	decrease of 1.0 kg	remained stable	
Shah et al. [74]	kidney	24	Canagliflozin	decrease from 8.5 ± 1.5% to 7.6 ± 1%	no data	decrease from 78.6 ± 12.1 kg to 76.1 ± 11.2 kg after 6 months (*p* < 0.05)	decrease from 86 ± 20 to 83 ± 18 mL/min per 1.73 m^2^ (*p* > 0.05)	
Attallah et al. [75]	kidney	8	Empagliflozin	decreased by 0.85 g/dL	no data	decrease of 2.4 kg after 1 year	no data	
**GLP-1**								
Pinelli et al. [76]	kidney	5	Liraglutide	no data	no data	decrease of 2.1 ± 1.3 kg after 21 days of treatment	no data	There were no differences in fasting blood glucose levels (5.0 ± 1.2 vs. 5.3 ± 0.5 mmol/L). Liraglutide seemed to reduce blood glucose levels at 60 (7.3 ± 1.2 vs. 5.9 ± 0.5 mmol/L) and 120 min (7.1 ± 0.8 vs. 6.0 ± 0.4 mmol/L).
Liou et al. [70]	kidney	7	Liraglutide	decreased from 10.0 ± 1.6% to 8.1 ± 0.8% (*p* = 0.043)	no data	decrease from 78.0 ± 7.8 kg to 75.1 ± 9.1 kg (*p* = 0.032)	increase from 67.7 ± 18.7 to 76.5 ± 18.7 mg/dL (*p* = 0.024)	Glycaemic control improved fasting blood sugar (FBS) from an initial level of 228.6 ± 39.1 mg/dL to a final FBS of 166.0 ± 26.6 mg/dL (*p* = 0.103), with a significant improvement in lowest glucose control 136.4 ± 5.8 mg/dL, (*p* = 0.017).
Singh et al. [77]	kidney, liver, heart	63	Dulaglutide	decreased by 10%, 5.3% and 8.4% after 6, 12 and 24 months	no data	decrease of 2%, 4% and 5.2% after 6, 12 and 24 months	increase of 15% after 24 months	
Singh et al. [77]	kidney, liver, heart	25	Liraglutide	initially decreased by 5.3% and 3% after 6 and 12 months, followed by an increase of 2% after 24 months	no data	decrease of 0.09%, 0.87% and 0.89% after 6, 12 and 24 months	decrease of 8% after 24 months	
Kukla et al. [78]	14 (82%) kidney, 1 (5.9%) kidney and liver, 2 (11.8%) kidney and heart	17	Liraglutide, Exenaglutide, Dulaglutide	remained statistically constant	no data	There was no statistical difference; only in 7 patients there was a decrease of 8.6 kg after 12 months (*p* = 0.07)	remained stable	Fasting blood glucose remained statistically unchanged. There was a significant reduction in total daily insulin dosage by a median of 30 IU (*p* = 0.007).
Thangavelu et al. [79]	7 kidney, 7 liver, 5 heart	19	Exenaglutide, Liraglutide, Dulaglutide, Semaglutide	decreased by 1.08%, 0.96% and 0.75% at 3, 6 and 12 months	no data	decrease of 4.86 kg after 12 months	remained stable after 12 months	

## Data Availability

No data were used for the research described in the article.

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
