# Peer review of "Managing Post-Transplant Diabetes Mellitus after Kidney Transplantation: Challenges and Advances in Treatment"

_pharmaceuticals, 2024, doi:10.3390/ph17080987_

Round 1

Reviewer 1 Report

Comments and Suggestions for Authors

The significant risk of developing post-transplantation diabetes mellitus in the kidneys as well as other organs and tissues makes the article relevant. This risk is mostly brought on by the continuous immunosuppressive therapy and the subsequent toxic effects of medications on pancreatic beta-cells. The lack of precise criteria for identifying the onset of posttransplantation diabetes mellitus, patient management strategies, and therapeutic approaches that would lessen the risk of developing such a severe complication also contributes to the necessity for such an article.

A notable aspect of the work is the systematic and comprehensive analysis of the effects of drugs used in kidney transplant patients by Grzegorz Rudzki et al. Managing Post-Transplant Diabetes Mellitus After Kidney Transplantation: Challenges and Advances in Treatment. This analysis included a review of the most commonly used drugs and their side effects, as well as information about new and promising drug substances that may be further explored for the treatment of these patients.

The lack of a methodical approach to writing a review article is the paper's main flaw. The authors also fail to disclose information regarding the timing of the analysis of the articles (i.e., when the articles were taken into consideration), how to search for papers in scientific article databases (PubMed, PRISMA), inclusion and exclusion criteria, etc. The article's structure also suggests that it is an original work rather than a review; the introduction and the chapter on the issues surrounding posttransplantation diabetes mellitus are excellent, but they come after the discussion section. It would have been more logical to continue without the table of contents or to combine the following chapters under the heading "Basic drugs in the posttransplantation period and in the development of posttransplantation diabetes mellitus."

Because of this, the review article's structure is a little unclear, but overall, the work is solid and provides a clear understanding of the major medications and their mechanisms of action, which can be helpful for practicing physicians and others looking to link their work to this area of medicine.

Notably, 35 review publications from 2019 through 2024 that cover different facets of post-transplant diabetes mellitus may be found on PubMed. As an illustration: Lawrence SE, Chandran MM, Park JM, Sweiss H, Jensen T, Choksi P, Crowther B. Sweet and simple as syrup: A review and guidance for use of novel antihyperglycemic agents for post-transplant diabetes mellitus and type 2 diabetes mellitus after kidney transplantation. Clin Transplant. 2023 Mar;37(3):e14922. doi: 10.1111/ctr.14922. Epub 2023 Feb 13. PMID: 36708369.

Hecking M, Sharif A, Eller K, Jenssen T. Management of post-transplant diabetes: immunosuppression, early prevention, and novel antidiabetics. Transpl Int. 2021 Jan;34(1):27-48. doi: 10.1111/tri.13783. Epub 2020 Nov 28. PMID: 33135259; PMCID: PMC7839745.

Alajous S, Budhiraja P. New-Onset Diabetes Mellitus after Kidney Transplantation. J Clin Med. 2024 Mar 27;13(7):1928. doi: 10.3390/jcm13071928. PMID: 38610694; PMCID: PMC11012473.

Rodríguez-Rodríguez AE, Porrini E, Hornum M, Donate-Correa J, Morales-Febles R, Khemlani Ramchand S, Molina Lima MX, Torres A. Post-Transplant Diabetes Mellitus and Prediabetes in Renal Transplant Recipients: An Update. Nephron. 2021;145(4):317-329. doi: 10.1159/000514288. Epub 2021 Apr 26. PMID: 33902027. However, this does not lessen the review article's significance; in fact, the more of these kinds of works, the better.

64% of the time, the paper incorporated pertinent, problem-related items that were no more than five years old at the time of publication. No indications of excessive self-citation are seen.
The authors use primary sources to support their summary of the significance of the factors including the incidence of posttransplantation diabetes mellitus, the lack of clear criteria for its detection, treatment options, and prevention of this complication in the article's final section.

It is easier to comprehend visually the role of risk factors for the development of posttransplantation diabetes mellitus and the effects of antidiabetic medicines when there are tables summarizing the findings of studies on these topics.

Some errors were found when this work was analyzed. For example, in chapter 2.1 (page 2, line 13–14), at the end of the sentence, "or shortened blood cell survival in transplant patients," could you please clarify if you're referring to shortened survival of red blood cells or all peripheral blood cells?

Page 3 - The main pathophysiological causes of PTDM development are insulin resistance and decreased pancreatic β-cell function, to which certain risk factors predispose. bold Is this emphasized for the readers by the authors, or is it a subsection?

The ordinal numbers of the cited literature in Tables 1 and 2 do not need to be placed in the title because they were already mentioned when the tables were initially mentioned.

Table 2: Although references 69–77 are listed before it in the text, does the table itself actually contain 68–77?

It is desirable to give a decoding of abbreviations, as not all readers are specialists in molecular biology.

Punctuation should be carefully checked by authors because occasionally words and references are combined, dots should appear before references, and Latin should be written in italics. 

Author Response

Comments 1:
The lack of a methodical approach to writing a review article is the paper's main flaw. The authors also fail to disclose information regarding the timing of the analysis of the articles (i.e., when the articles were taken into consideration), how to search for papers in scientific article databases (PubMed, PRISMA), inclusion and exclusion criteria, etc. The article's structure also suggests that it is an original work rather than a review; the introduction and the chapter on the issues surrounding posttransplantation diabetes mellitus are excellent, but they come after the discussion section. It would have been more logical to continue without the table of contents or to combine the following chapters under the heading "Basic drugs in the posttransplantation period and in the development of posttransplantation diabetes mellitus."

Response 1:
Thank you for highlighting the issue. We have divided the paper as suggested and added a new paragraph 'Methods' including information about the timing of the analysis of the articles, how search for papers in scientific article databases and inclusion and exclusion criteria.
Excerpt attached in the article:
“Methods

The systematic review aims to summarize the latest literature regarding the latest treatment of diabetes in renal transplant patients. Current literature was reviewed by searching for publications from the period between 2014 and 2024 using a search phrase “New Onset Diabetes After Transplantation” AND “Post-Transplant Diabetes Mellitus” AND “kidney transplantation” in Pubmed database. Three authors independently assessed the eligibility and quality of the study and performed data extraction. All selected studies were included if the study population consisted of adult renal transplant patients diagnosed with impaired glucose tolerance, diabetes mellitus (DM), new onset post-transplant diabetes (NODAT) or post-transplant diabetes mellitus (PTDM). The exclusion criteria were as follows: articles available in a language other than Polish or English, as well as other publications types as editorials, reviews, posters and letters.”
In the revised version of the manuscript, the change is on page 4, chapter 2 and line from 119 to 130.

Comments 2:
Some errors were found when this work was analyzed. For example, in chapter 2.1 (page 2, line 13–14), at the end of the sentence, "or shortened blood cell survival in transplant patients," could you please clarify if you're referring to shortened survival of red blood cells or all peripheral blood cells?

Response 2:
In chapter 2.1 it referred to the red blood cells. We have modified this statement. In the revised version of the manuscript, the change is on page 2, chapter 1 and line 55.

Comments 3:
Page 3 - The main pathophysiological causes of PTDM development are insulin resistance and decreased pancreatic β-cell function, to which certain risk factors predispose. bold Is this emphasized for the readers by the authors, or is it a subsection?

Response 3:
Thank you for your comment bold was used by mistake. A correction has been made in the manuscript

Comments 4:
The ordinal numbers of the cited literature in Tables 1 and 2 do not need to be placed in the title because they were already mentioned when the tables were initially mentioned.

Response 4: 
Thank you for your suggestions. We have corrected and removed citations from the titles of tables. In the revised version of the manuscript, the change is on page 3, chapter 1 and line 115 (Table 1); page 9, chapter 3 and line from 387 to 388 (Table 2).

Comments 5:
Table 2: Although references 69–77 are listed before it in the text, does the table itself actually contain 68–77?

Response 5:
We agree with this comment. There was an mistake related to the bibliography. The actual data in the table are for items 70-79, which has been corrected in the text. The changes in the bibliography are due to the addition of 2 new citations in the text of the manuscript before the table.
In the revised version of the manuscript, the change is on page 8, chapter 3 and line 385.

Comments 6:
It is desirable to give a decoding of abbreviations, as not all readers are specialists in molecular biology. Punctuation should be carefully checked by authors because occasionally words and references are combined, dots should appear before references, and Latin should be written in italics. 

Response 6:
Thank you for taking note. We have applied the suggested changes to our manuscript. The article was rechecked, punctuation corrections were made, missing abbreviations were developed, and reviewer's comments included.

Reviewer 2 Report

Comments and Suggestions for Authors

The authors have provided a useful article in Managing Post-Transplant Diabetes Mellitus After Kidney Transplantation: Challenges and Advances in Treatment.

The title is appropriate and it is in the scope of the journal.
The abstract of the article needs to be revised and the  method and conclusion sections should be rewritten in that part.
The introduction is very short and includes two references, which is not acceptable in any scientific article.
Table 1 is not informative.
The need for pictures and diagrams is felt in the method section.
Table 2 is long and not uniform. It is suggested to categorize and summarize.
In general, the article does not have enough order and coherence, and its process is not communicative or evolutionary.

Author Response

Comments 1:
The abstract of the article needs to be revised and the method and conclusion sections should be rewritten in that part.

Response 1:
Thank you for highlighting the issue. Due to the limited word count of 200 words, we are unable to include a methods section in the abstract. We have added a new paragraph 'Methods' into main part of the article (chapter 2, page 4, line from 119 to 130), including information about the timing of the analysis of the articles, how search for papers in scientific article databases and inclusion and exclusion criteria.

Comments 2:
The introduction is very short and includes two references, which is not acceptable in any scientific article.

Response 2:
Thank you for your valuable consideration. We have expanded our manuscript and rearranged our paper. The introduction is now comprehensive and includes a wide range of references.

Comments 3:
Table 2 is long and not uniform. It is suggested to categorize and summarize.

Response 3:
We have tried to make the table a summary of data from various studies on PTDM treatment and allow a collective evaluation of the parameters compared. The issue of inconsistency of data in the table, may be due to the fact that the results were obtained from completely unrelated studies.  The issues discussed in the table relate to the main text of the article, so again, we would not like to duplicate them in the content.

Reviewer 3 Report

Comments and Suggestions for Authors

The provided abstract for the review study "Managing Post-Transplant Diabetes Mellitus After Kidney Transplantation: Challenges and Advances in Treatment," the following suggestions must be addressed to improve the study.

The abstract should provide any quantitative information about the prevalence of PTDM or its impact on patient outcomes.

The phrase "quality of life and survival of both patients and the transplanted organ" is unclear. Clarify it

The title mentions "Challenges," the abstract doesn't clearly outline what these challenges are in managing PTDM.

Some drug classes are named (e.g., DPP-4 and SGLT-2 inhibitors), while others are referred to by specific drug names (e.g., dorzagliatin). A consistent approach would be better for clarity.

In introduction include statistics on the prevalence of PTDM among kidney transplant recipients.

Briefly mention the main risk factors for developing PTDM, such as immunosuppressive medications, age, obesity, etc.

Discuss when PTDM typically develops post-transplantation, as this can be important for monitoring and early intervention.

Highlight some of the unique challenges in managing diabetes in transplant patients, such as drug interactions or the impact on immunosuppression.

Mention if there are any specific guidelines for managing PTDM and how they might differ from standard diabetes management

The introduction cites two references. Add some recent studies such as http://doi.org/10.36899/JAPS.2022.3.0484, https://doi.org/10.3390/molecules27196728

Glucose levels are given in both mg/dL and mmol/L, but the conversion is not consistent throughout. For example, "200 mg/dL (11.1 mmol/L)" is correct, but "126 mg/dL (7 mmol/L)" should be "126 mg/dL (7.0 mmol/L)" for consistency

Commoditized hyperglycemic symptoms" should likely be "concomitant hyperglycemic symptoms.

The sentence "The late onset is comparable to the incidence of classic DMT2" is unclear. It's not specified what "late onset" refers to or how it compares to DMT2 incidence

The sentence "The impact of timing on the diagnosis of post-transplant diabetes has still to be established" seems to contradict earlier statements about PTDM developing most frequently between 3 and 6 months post-transplant.

The text uses both "DMT2" and "type 2 diabetes" interchangeably. It would be better to stick to one term for consistency.

The section on amycretin acknowledges the limited available information but doesn't specify which phase of clinical trials it's currently in.

This study should provide more specific recommendations or conclusions about the potential place of these new drugs in diabetes management

Author Response

Comments 1:
The abstract should provide any quantitative information about the prevalence of PTDM or its impact on patient outcomes.

Response 1:
We have added your suggestion, which is: “The prevalence of PTDM, depending on the source, can range from 4 to 30% of transplant patients.” Unfortunately, for the limited amount to 200 words in the abstract, we could not include more additional information. In the revised version of the manuscript, the change is abstract, on page 1 and line from 20 to 21.

Comments 2:
The phrase "quality of life and survival of both patients and the transplanted organ" is unclear. Clarify it

Response 2:
Thank you for your valuable feedback on the article. Regarding your second point, it is important to note that post-transplant diabetes mellitus (PTDM) can exacerbate other complications such as infections and acute rejection episodes, further jeopardizing graft survival. The primary focus of the paper is on hypoglycemic medications rather than the quality of life aspects. Your insights will be duly considered for enhancing the clarity and relevance of the manuscript.

We have added your suggestion, which is: “Implying complications including infections and episodes of acute rejection, further threatening graft survival.” In the revised version of the manuscript, the change is abstract, on page 1 and line from 19 to 20. Unfortunately, for the limited amount to 200 words in the abstract, we could not include more additional information.

Comments 3:
The title mentions "Challenges," the abstract doesn't clearly outline what these challenges are in managing PTDM.

Response 3:
In response to your third point, it is worth mentioning that PTDM still remains a carbohydrate disorder that is not well understood and may pose both diagnostic and therapeutic challenges for practicing physicians. Additionally, the timing of PTDM diagnosis and the lack of globally established treatment algorithms are challenges as highlighted in the title of the article. These aspects will be further emphasized and elucidated in the revised manuscript to provide a more comprehensive overview. Your input is invaluable in ensuring the robustness and relevance of the study.

We have added your suggestion, which is: “Although many therapeutic options are currently available, PTDM often creates uncertainty about the most appropriate treatment strategy. Therefore, more research is needed to individualize therapeutic plans and monitor these patients “
In the revised version of the manuscript, the change is abstract, on page 1 and line from 28 to 31. Unfortunately, for the limited amount to 200 words in the abstract, we could not include more additional information.

Comments 4:
Some drug classes are named (e.g., DPP-4 and SGLT-2 inhibitors), while others are referred to by specific drug names (e.g., dorzagliatin).

Response 4:
A consistent approach would be better for clarity. The consistency in naming drug classes and specific drug names has been duly rectified in the revised version of the manuscript.

Comments 5:
In introduction include statistics on the prevalence of PTDM among kidney transplant recipients. Briefly mention the main risk factors for developing PTDM, such as immunosuppressive medications, age, obesity, etc. Discuss when PTDM typically develops post-transplantation, as this can be important for monitoring and early intervention. Highlight some of the unique challenges in managing diabetes in transplant patients, such as drug interactions or the impact on immunosuppression. Mention if there are any specific guidelines for managing PTDM and how they might differ from standard diabetes management

Response 5:
Thank you for your valuable consideration. Information such as statistics on the prevalence of PTDM, risk factors for developing PTDM and its pathomechanism were covered extensively in a later section of our manuscript, but following this comment, we have rearranged our paper and combined those information as an entire single 'introduction' section.

Comments 6:
The introduction cites two references. Add some recent studies such as http://doi.org/10.36899/JAPS.2022.3.0484, https://doi.org/10.3390/molecules27196728

Response 6:
Thank you for pointing this out. We rearranged our paper. The manuscript is now comprehensive and includes a wide range of references. In the revised version of the manuscript, the change on page 6 and line from 227 to 233.

Comments 7:
Glucose levels are given in both mg/dL and mmol/L, but the conversion is not consistent throughout. For example, "200 mg/dL (11.1 mmol/L)" is correct, but "126 mg/dL (7 mmol/L)" should be "126 mg/dL (7.0 mmol/L)" for consistency .

Response 7:
Thank you for pointing this out. We agree with this comment. In the revised version of the manuscript, the change is to page 1 and 2, chapter 1 and line 46 and 49

Comments 8:
Commoditized hyperglycemic symptoms" should likely be "concomitant hyperglycemic symptoms
.

Response 8:
A revision has been made. In the corrected version of the manuscript, the change is to page 2, chapter 1 and line 47.

Comments 9:
The sentence "The late onset is comparable to the incidence of classic DMT2" is unclear. It's not specified what "late onset" refers to or how it compares to DMT2 incidence

Response 9:
Thank you for drawing attention to the ambiguity in the sentence "The late onset is comparable to the incidence of classic DMT2." I would like to explain what "late onset" means in the context of post-transplant diabetes mellitus (PTDM) and how it compares to the frequency of classic type 2 diabetes (DMT2). In the context of PTDM, "late onset" refers to cases of diabetes that develop after the first year following kidney transplantation. In medical literature, PTDM is often classified as early onset when it develops within the first 3-6 months post-transplantation and late onset when it develops after 12 months from the transplant. The incidence of late-onset PTDM is comparable to the incidence of classic type 2 diabetes (DMT2) in the general population. Studies indicate that the annual incidence of late-onset PTDM is around 7%, which aligns with the prevalence of DMT2 in the general population. In an article published in the "International Journal of Molecular Sciences," Rysz et al. (2021) point out that late-onset PTDM develops at a rate similar to the prevalence of DMT2 in the general population, approximately 7% annually. Other studies also confirm that the incidence of late-onset PTDM is similar to that of DMT2, suggesting that the pathophysiological mechanisms may be comparable, even though PTDM is triggered by additional factors related to transplantation, such as immunosuppression. I hope this explanation clears up any uncertainties regarding this statement.
In the revised version of the manuscript, the change on page 2 and line from 77 to 85.

Comments 10:
The sentence "The impact of timing on the diagnosis of post-transplant diabetes has still to be established" seems to contradict earlier statements about PTDM developing most frequently between 3 and 6 months post-transplant.

Response 10:
Thank you for pointing this out. I would like to explain this sentence. Studies indicate that PTDM typically develops within the first 3-6 months after kidney transplantation. This early post-operative period is critical due to factors such as surgical stress, high doses of corticosteroids, and the initiation of immunosuppressive therapies like calcineurin inhibitors, which contribute to hyperglycemia and the onset of diabetes.

The late onset of PTDM, when it develops after 12 months from the transplant, is comparable to the incidence of classic DMT2 in the general population. In an article published in the "International Journal of Molecular Sciences," Rysz et al. (2021) point out that late-onset PTDM develops at a rate similar to the prevalence of DMT2 in the general population, approximately 7% annually.
Therefore, we would like to emphasise that there is no cut-off period for the diagnosis of PTDM, which poses a diagnostic challenge.
We rearranged our paper and we hope this explanation clears up any uncertainties.

Comments 11:
The text uses both "DMT2" and "type 2 diabetes" interchangeably. It would be better to stick to one term for consistency.

Response 11: Updated text in the manuscript

Comments 12:
The section on amycretin acknowledges the limited available information but doesn't specify which phase of clinical trials it's currently in.

Response 12:
Thank you for pointing this out. We have applied the suggested changes to our manuscript and added 3 new citations (numbers:102-104).

We have added your suggestion, which is: “This drug has just completed phase 1 of clinical trials, and the company plans to initiate phase 2 in the second half of 2024. The preliminary study results are very encouraging. During the phase 1 clinical trials, participants lost an average of 13.1% of their body weight after 12 weeks of drug use.”

In the revised version of the manuscript, the change is to page 13 chapter 3.3.3.  and line from 501 to 503.

Comments 13:
This study should provide more specific recommendations or conclusions about the potential place of these new drugs in diabetes management

Response 13:
We have applied the suggested changes to our manuscript and add “Incorporating these new drugs into diabetes management protocols could offer novel therapeutic options for patients with diabetes, addressing specific aspects of the condition such as glucose control, weight management, and potential renal benefits. Continued research and clinical evaluation are essential to ascertain their efficacy, safety profiles, and optimal placement within diabetes treatment strategies” into conclusions.
In the revised version of the manuscript, the change is to page 14 chapter 4  and line from 549 to 553